# Applicability of Point- and Polygon-Based Vegetation Monitoring Data to Identify Soil, Hydrological and Climatic Driving Forces of Biological Invasions—A Case Study of *Ailanthus altissima*, *Elaeagnus angustifolia* and *Robinia pseudoacacia*

**DOI:** 10.3390/plants12040855

**Published:** 2023-02-14

**Authors:** Georgina Veronika Visztra, Kata Frei, Alida Anna Hábenczyus, Anna Soóky, Zoltán Bátori, Annamária Laborczi, Nándor Csikós, Gábor Szatmári, Péter Szilassi

**Affiliations:** 1Department of Physical Geography and Geoinformatics, University of Szeged, Egyetem utca 2, H-6722 Szeged, Hungary; 2Department of Ecology, University of Szeged, Közép fasor 52, H-6726 Szeged, Hungary; 3Department of Soil Mapping and Environmental Informatics, Institute for Soil Sciences, Centre for Agricultural Research, H-1022 Budapest, Hungary

**Keywords:** invasive tree species, LUCAS, forest units, ArcGIS, biological invasion, *Ailanthus altissima*, *Elaeagnus angustifolia*, *Robinia pseudoacacia*

## Abstract

Invasive tree species are a significant threat to native flora. They modify the environment with their allelopathic substances and inhibit the growth of native species by shading, thus reducing diversity. The most effective way to control invasive plants is to prevent their spread which requires identifying the environmental parameters promoting it. Since there are several types of invasive plant databases available, determining which database type is the most relevant for investigating the occurrence of alien plants is of great importance. In this study, we compared the efficiency and reliability of point-based (EUROSTAT Land Use and Coverage Area Frame Survey (LUCAS)) and polygon-based (National Forestry Database (NFD)) databases using geostatistical methods in ArcGIS software. We also investigated the occurrence of three invasive tree species (*Ailanthus altissima*, *Elaeagnus angustifolia*, and *Robinia pseudoacacia*) and their relationships with soil, hydrological, and climatic parameters such as soil organic matter content, pH, calcium carbonate content, rooting depth, water-holding capacity, distance from the nearest surface water, groundwater depth, mean annual temperature, and mean annual precipitation with generalized linear models in R-studio software. Our results show that the invasion levels of the tree species under study are generally over-represented in the LUCAS point-based vegetation maps, and the point-based database requires a dataset with a larger number of samples to be reliable. Regarding the polygon-based database, we found that the occurrence of the invasive species is generally related to the investigated soil and hydrological and climatic factors.

## 1. Introduction

Mapping and predicting the occurrence and potential distribution of invasive plants are of global significance. Alien species pose a heavy burden to natural ecosystems, displacing native species and transforming natural communities [1,2]. One of the main issues conservationists face is biological invasion [3]. In addition to natural areas, invasive plants also invade agricultural [4] areas and have a dramatic impact on urban areas [5,6], generating enormous extra costs to national economies all over the world [7].

Early detection and monitoring of existing populations of invasive species, and the prediction of areas potentially exposed to their expansion are the most important steps of a successful invasion control strategy [8,9,10]. To understand the environmental parameters (soil, hydrological, and climatic characteristics) determining the distribution of these species, and to model and predict their future spread, up-to-date and detailed spatial data are necessary [11,12].

Tree of heaven (*Ailanthus altissima*), Russian olive (*Elaeagnus angustifolia*), and black locust (*Robinia pseudoacacia*) show a rapid spread in Eurasian countries. Biological invasion is a very complex phenomenon, driven by various geographical factors (e.g., soil, hydrological, and climatic conditions; traffic, railway, and ecological networks; and land use change), and thus its comprehensive understanding requires a holistic approach that also considers geographical aspects [13,14,15,16]. To map and predict the potential habitats of invasive plant species, we need to know the relationships between geographical factors (covariates) and their impact on plant occurrence [10,14,15,17,18]. The spatial and thematic accuracy of habitat modeling for invasive species depends to a large extent on the spatial accuracy of botanical surveys and other (soil, hydrological, climatic) input databases used for modeling. To be able to use these databases for modeling hazard mapping, we need to determine which types of database results provide the most reliable information. Currently, point maps based on citizen science-based data collection of invasive plants are increasingly used to explore the environmental context of biological invasions. However, the scientific utility of these so-called fragmented big databases (such as Global Biodiversity Information Facility and iNaturalist) is severely limited by the fact that they provide only point information on the occurrence of the plants under study, their sample points are not uniformly distributed, and they do not provide any further information on points where a certain plant species does not occur [19]. To remedy this problem, we constructed country-scale point occurrence maps, known as the National GIS Database of Invasive Plant Species of Hungary (NDIPS), of the three investigated invasive tree species, showing the status of invasions for each Land Use and Coverage Area frame Survey (LUCAS) field survey point (non-invaded: no invasive species visible; invaded: at least some individual plants of a given species are visible on the LUCAS photos). For our analyses, LUCAS points that do not contain invasive plants also provide useful information due to the uniform point distribution of the LUCAS data obtained from LUCAS photos [14,15].

In this study, we investigated the occurrence of three invasive tree species (*Ailanthus altissima*, *Elaeagnus angustifolia,* and *Robinia pseudoacacia*). Among woody plants, these species present the highest risk of biological invasion in Hungary [20,21]. 

*A. altissima* has very low ecological requirements, and it can grow on low-productivity, nutrient-poor debris soils; for that reason, it is often planted in cities to absorb carbon dioxide and airborne particulate matter [6]. Because of its low habitat requirements, it can even be found in pavement cracks in cities [22]. In addition, previous research confirms that in the large Polish city of Wrocław, *A. altissima* is concentrated on urban heat islands where the air temperature is a few degrees higher [6]. However, it spreads easily from urban areas and invades other disturbed vegetation and natural forests without a closed canopy [23]. Furthermore, *A. altissima* produces allelopathic substances that inhibit the growth of other plants, and its shading effect is also significant [24,25,26]. *E. angustifolia* is mainly used for the afforestation of saline and debris soils because its soil requirements are extremely undemanding. In addition, it has been used extensively in field protection forest strips [23]. In treeless habitats, it overshadows light-demanding species and causes the decline of many rare and protected plants. *E. angustifolia* lives in a symbiotic relationship with nitrogen-fixing bacteria, thus promoting nitrogen accumulation in the soil, which promotes the establishment and emergence of nitrogen-favoring weed species [23]. In addition, similar to *A. altissima, R. pseudoacacia* produces allelopathic substances that transform the abiotic conditions of the habitat, thus displacing native species from the surrounding areas [24]. All three species are, therefore, major contributors to the transformation of their environment, leading to a loss of diversity of native species [3,23,24,26,27]. In many cases, mapping the occurrence of invasive species is very expensive and requires much work, and as a consequence, such research attempts are rather limited. Paź-Dyderska et al. (2020) suggested applying land use maps and databases of invasive plants to reduce costs [5]. According to this suggestion, we used different qualitative and quantitative geographical databases to identify the main geographical (climatic, hydrological, and soil) factors (driving forces) that determine the occurrence of alien plants in Hungary [14,15]. 

In this study we sought to answer the following research questions: What is the reliability of different data sources (point- or polygon-based vegetation datasets) for the monitoring of biological invasion? What soil, hydrological, and climatic parameters influence the occurrence of the considered invasive species?How different are the results obtained by comparing point and polygon-based vegetation databases and environmental data?

## 2. Results

### 2.1. Comparing the Reliability of Point and Polygon-Based Invasion Maps

The number of invaded LUCAS points was the highest for *R. pseudoacacia*, whereas *E. angustifolia* was detected in the lowest number of LUCAS points (Table 1).

Compared to the 2021 NFD statistics used as reference data, the level of invasion of the investigated tree species was generally over-represented on the LUCAS point-based vegetation maps. The smallest difference (only approximately 7%) was found for the tree species with the largest proportion of area (*R. pseudoacacia*), while *E. angustifolia* was 30 times over-represented in the invaded LUCAS points identified from field photographs, and *A. altissima* was over-represented by more than 100 times compared to the actual reference data (Table 2). 

The area covered by *R. pseudoacacia* was underestimated by 9% in the total forested area, while the area covered by *E. angustifolia* was underestimated by 0.04% and the area covered by *A. altissima* was underestimated by 0.9% in the polygonal occurrence map compared to the NFD statistics reference data. Thus, our results show that all polygon-based vegetation maps underestimate the area invaded by all species.

A comparison of point and polygon-based invasion maps shows that the LUCAS points invaded by *R. pseudoacacia* coincide with the polygons most invaded by this species (Appendix B).

In the case of *A. altissima* and *E. angustifolia*, the LUCAS points invaded by the species were also found within polygons (forested areas) not invaded by the species. This finding suggests that individuals of all investigated invasive tree species are often mixed with other species and occur along forested edges or in patches along roadsides and channels. 

### 2.2. Database-Source Discrepancies in the Relationship between Environmental Variables and the Occurrence Maps of the Investigated Invasive Species

We observed significant differences in environmental variables between invaded and non-invaded areas. For *A. altissima*, we found differences in the organic matter content of different soil layers. Examining the LUCAS points, the organic matter content was significantly lower in the 0–30 and 30–60 cm soil layers at the invaded points than at the non-invaded points. In the case of NFD polygons, (except for the 0–30 cm soil layer), the organic matter content was significantly higher at the invaded polygons. Soil pH was consistently significantly higher in invaded than non-invaded areas based on both databases.

Considering the LUCAS points, we found no significant difference in calcium carbonate content; however, for the NFD polygons, calcium carbonate content was significantly higher at the invaded polygons in the case of *A. altissima*. Rooting depth was significantly higher at the invaded LUCAS points, but we found no significant difference at the NFD polygons. 

In both databases, the water-holding capacity of the soil in the 0–30 and 30–60 cm layers was significantly lower in the case of *A. altissima*. Regarding the distance from surface water, we found contrasting results. Based on the LUCAS database, distances from surface water were significantly higher in the invaded areas; however, based on the NFD polygon database, distances from surface water were significantly lower in the invaded areas. Significant differences in groundwater depth were found only when NFD polygons were considered. At the polygons invaded by *A. altissima*, groundwater depth was significantly higher than at the non-invaded polygons. The mean annual temperature was significantly higher and the mean annual precipitation was significantly lower in the case of invasion, considering both databases (Table 3). 

Regarding *E. angustifolia*, we found very different results considering the LUCAS and NFD polygon databases. For the organic matter content of the soil, we found significant differences in the 30–60 cm and 60–100 cm soil layers at the invaded LUCAS points. However, considering the NFD polygons, the organic matter content was significantly higher in all soil layers invaded by *E. angustifolia*. The soil pH was also significantly higher at the invaded LUCAS points and NFD polygons. The calcium carbonate content was significantly higher at the invaded NFD polygons; however, we found no significant differences in the 100–200 cm soil layer between invaded and non-invaded LUCAS points.

There were no significant differences in rooting depth between invaded and non-invaded LUCAS points. However, rooting depth was significantly lower in the invaded NFD polygons. In the case of water-holding capacity, we found no significant differences between invaded and non-invaded LUCAS points, but when considering the NFD polygon database, we found significant differences between invaded and non-invaded polygons. In the 0–30 cm soil layer, the water-holding capacity was significantly lower, whereas in the 30–60, 60–100, and 100–200 cm soil layers, the water-holding capacity was significantly higher at the invaded polygons. Distance from surface water did not show a significant difference considering either database. Groundwater depth was significantly lower and mean annual temperature was significantly higher in the case of invasion with both databases. The mean annual precipitation was significantly lower in the invaded areas in the polygon database, but we did not find a significant difference in the point database (Table 3).

Regarding *R. pseudoacacia,* significant differences were found between invaded and non-invaded areas using both LUCAS points and NFD polygons. In the case of the NFD polygon database, we found significant differences in all the soil and climatic conditions (except in the calcium carbonate content in the 60–100 soil layer). The same was true in the case of the LUCAS point database, except for the calcium carbonate content in the 30–60 and 60–100 cm soil layers and the water-holding capacity in the 30–60 cm soil layer. The organic matter content was significantly lower in the case of invasion in all soil layers according to the LUCAS database and in all soil layers to the NFD polygon database except for the 100–200 cm soil layer. In the 100–200 cm soil layer, the organic matter content was significantly higher at the invaded polygons. When there was an invasion by *R. pseudoacacia*, soil pH was significantly higher in both databases. The calcium carbonate content was significantly lower at the invaded points and polygons in the 0–30 and 100–200 cm soil layers, and in the case of NFD polygons, the calcium carbonate content was significantly lower in the 30–60 cm soil layer as well. Rooting depth and mean annual temperature were significantly higher, and groundwater depth and mean annual precipitation were significantly lower when the invasion was present in both databases. The water-holding capacity in the 0–30 cm soil layer was significantly lower in the case of invasion in both databases. In the other investigated soil layers, the water-holding capacity was significantly higher (except for 30–60 cm in the LUCAS database, where we did not find any significant difference) (Table 3).

## 3. Materials and Methods

### 3.1. Study Area

The forested areas (i.e., semi-natural forests, secondary forests, and plantations) comprise approximately 22.8% of the land in Hungary, the majority of which (64%) is semi-natural temperate deciduous forest (forests dominated by *Quercus petraea*, *Q. cerris*, *Carpinus betulus*, and/or *Fagus sylvatica*) [28,29]. Of the total forested area in Hungary, 11% is covered by pine plantations, and 25% is planted with *Robinia pseudoacacia*. According to 2021 data, *Ailanthus altissima* covered 0.11% of the total forested area in Hungary, while *Elaeagnus angustifolia* covered 0.08% (National Forestry Database 2021) [30]. Continuous forest cover is mainly found in the mountainous and hilly areas of the country, while in the Great Hungarian Plain, riparian forests are dominant along rivers (Figure 1), although sandy areas give rise to an unaccounted percentage of non-native plantations.

The climate of the Hungarian forests is wet–temperate with a mean annual temperature of 8.0–10.5 °C; and 500–700 mm average yearly precipitation [31]. The main soil types in hilly forest areas are brown forest soils, while rendzina soils are present in limestone mountain areas, and ranker soils are present in volcanic and metamorphic areas [32]. 

### 3.2. Digital Databases 

For the analysis of the spatial distribution (occurrence) of the three investigated invasive tree species within Hungarian forest areas we used point- and forest-unit (polygon)-based digital maps. Both the point- and polygon-based databases of the different vegetation monitoring methods provide a picture of the invasion of the entire forest area of Hungary in 2012, 2015, and 2018. 

#### 3.2.1. Point-Based GIS Database of Invasive Plant Species of Hungary

In 2012, 2015, and 2018, 3432 geotagged points inside the forested areas of Hungary were stored in the EUROSTAT Land Use and Land Cover Survey (LUCAS) database, where the actual status and change in land use and land cover were determined based on three yearly field investigations. At each LUCAS point, five field geotagged photos were taken from the four cardinal directions (N, E, W, and S) and downwards from the point itself [14]. These GPS-recorded (geotagged) field photos from 2012, 2015, and 2018, offer unique possibilities to identify the level of invasion of each investigated tree species and to monitor the level of invasion in the forested areas of Hungary. A LUCAS point was considered to be invaded if we identified at least one individual of the invasive tree species in at least one of the field photographs. Following the visual interpretation of more than 18,000 field photographs, we produced a map showing each LUCAS point whether it was invaded by one of the three invasive tree species investigated (Figure 2). 

#### 3.2.2. Polygon-Based Forest Monitoring Data

We also used the field source data of the invasion surveys of the National Forestry Database (NFD) of Hungary on the three investigated invasive tree species. The forested area of Hungary is divided into a few hectares of territory, known as, forest units. The digital forest map of the NFD used for our research shows the percentage of invasion of a given species within the spatial units (polygons) of forested areas in Hungary, which represents only 21% of the whole country. Forest units that were at least 90% invaded at least once in 2012, 2015, and 2018, were considered invaded polygons, those with 0% invasion were considered non-invaded polygons (Figure 2), which were half-reduced by random selection for easier processing during the statistical analysis. 

In addition to the NFD polygon-based forest maps, we also used NFD statistical data as a reference for our research to represent the real proportions of invasion. The NFD statistics provide a reliable indication of the percentage of the total forested area in Hungary covered by the invasive tree species under study. These statistics are calculated on the basis of forest units with an invasion rate of more than 90% and represent the total area covered by all invasive tree species. 

For each of the three species, the invasion of forested areas was expressed as the percentage of LUCAS points within the total forested area that were invaded, and as the percentage of the total forested area covered by NFD forest units at least 90% invaded by the species. The results of these two calculations were compared with the reference NFD statistical data.

To test the spatial accuracy of the used datasets, the LUCAS point- and NFD polygon-based occurrence maps of the investigated invasive trees were merged in ArcGIS 10.7 software. The accuracy of the point and polygon-based invasion maps was tested by calculating the invasion values as a percentage of total LUCAS points for the point-based map and as a percentage of total forested area for the forest unit (polygon) map, and then comparing them with the species level land cover data (National Forestry Database 2021) [30], which can be used as reference data. 

The reliability of the databases was tested by calculating for each species how many invaded LUCAS points fall in the invaded or non-invaded NFD forest unit (polygon) of the given species and how many non-invaded LUCAS points fall in the invaded or non-invaded NFD forest units (polygon). Formulas were developed to carry out the calculations (Appendix A). 

#### 3.2.3. Soil, Hydrological, and Climate Databases Used

For the spatial characterization of the environmental (soil, hydrological, and climatic) variables used for the analysis, Digital, Optimized, Soil Related Maps and Information in Hungary (DOSoReMI.hu) datasets with resolutions of 100 × 100 m and 250 × 250 m were used.

The DOSoReMI digital soil database [33] was used to analyze the role of the soil covariates on occurrences of the three investigated tree invasions. As soil hydraulic properties, the saturated water content (THS), water content at field capacity (FC), and water content at wilting point (WP) values from the 3D Soil Hydraulic Database of Europe (EU-SoilHydroGrids ver1.0) were used [34]. The hydrological parameters taken into account were groundwater depth [35] and distance from the nearest surface water. The elevation and topography factors were derived from a publicly available digital elevation model with a spatial resolution of 30 m or better, while the long-term (1960–1990) climatic averages of Hungary were obtained from the Hungarian Meteorological Service dataset [36] (Table 4.).

### 3.3. Statistical Methods

#### GIS and Statistical Analysis

Using ArcGIS software 10.7, we overlaid the soil, hydrological, and climatic data (DOSoReMI database) with the invaded and non-invaded LUCAS points and NFD polygons. This method was used to determine the environmental parameters at the given points and polygons. A generalized linear model (GLM) was used to determine in which direction (positive or negative) and to what extent each parameter differs between invaded and non-invaded points and polygons. The p-value indicates whether the investigated factor differs between invaded and non-invaded areas; the z value indicates in which areas (invaded or non-invaded) the soil, hydrological, and climatic factor values are higher. If the *p-value* is <0.05, the factor has a significant impact on the occurrence of the given species. When the z value is <0, the given factor is higher at the non-invaded points and polygons, and if the z value is >0, the considered factor is higher at the invaded areas. For the GLM, we used the *glm*() function in R-studio software 4.0 [37].

## 4. Discussion

The relationship of the investigated plant species to the soil, hydrological, and climatic conditions investigated is highly variable. In a few cases, according to the LUCAS point database and NFD polygon database, we found different results. In general, when considering the NFD polygon database, we found more significant differences in the soil, hydrological, and climatic parameters than when considering the LUCAS point database. We found the most variation in the results in the case of *E. angustifolia*, where the invaded LUCAS points were the least variable. Most similarities were found in the case of *R. pseudoacacia*, for which the highest number of invaded points was observed. Consequently, the reliability of the point database increases with the amount of data and is only reliable if sufficient data are available.

Several studies note that the main weakness of crowd-sourced, geotagged, photo-based vegetation maps is that the spatial density of field data is very diverse, that is, very spatially fragmented data [38,39,40]. However, the LUCAS database is based on spatially uniform survey points with uniform data density, and therefore, the representativeness of the data is higher compared to crowd-sourced vegetation databases. The analysis of the LUCAS geotagged photos [14] together with other databases allowed us to compare the soil characteristics between the invaded and non-invaded LUCAS points. For all investigated tree species, the estimated invasion rates based on the ratio of invaded LUCAS points to the total number of LUCAS points within the forested areas were higher than the actual invasion rates of the species in the reference statistics for forested areas in Hungary. This can be explained by the fact that while the reference data showed the percentage cover (surface cover) of each invasive tree species, the geotagged photographs of the LUCAS-based database also identified many invaded points. At these points, the species occurrence is not homogeneous, for example, on the edges of forest patches dominated by other species, on roads, ditches, etc. Our results, therefore, suggest that point-based databases are a spatially more accurate representation of invasive species dispersal pathways than polygon (forest unit) patch-based vegetation databases [41]. In our previous study [15], we observed the largest difference between the spatial characteristics of invaded forest units (polygons) and invaded LUCAS points for *A. altissima* and *E. angustifolia* [15] spreading along roads and railways [15], which supports that point-based databases can be very useful for generating dispersal models. Conversely, the difference between the spatial pattern of homogeneous patches and point occurrences is smaller for *R. pseudoacacia*, as this species is planted over large areas in Hungary, regardless of its harmful effects on the native flora. 

Similar to other studies [13,14,15], we found that the investigated soil, hydrological, and climatic factors had a large impact on the occurrence of the studied invasive plant species. *A. altissima* was found mainly on degraded soils with a low organic matter content [23]. This does not necessarily mean that it prefers soils with low organic matter content, but in these areas, it has few or no competitors and thus can easily establish [42]. Our results confirm that *A. altissima* tolerates calcareous soils as well, which also have a higher pH due to the high calcium carbonate content. The rooting depth of *A. altissima* was found to be higher at the invaded LUCAS points than at the non-invaded LUCAS points. This is somewhat surprising, as the poor quality, debris-laden soils where *A. altissima* is commonly found only allow for a shallow rooting depth. The soils favored by *A. altissima* are characterized by a reduced water-holding capacity, and furthermore, the mean annual precipitation is significantly lower in its preferred habitats. It follows that this plant tolerates drought well and has good water absorption capacity [10,15]. As *A. altissima* prefers warmer and drier habitats, global warming may facilitate the spread of this alien plant in the future [43]. This finding is supported by the fact that the distribution of *A. altissima* in Wrocław, for instance, is concentrated on urban heat islands [6].

We found that *E. angustifolia* is most abundant in soils with high organic matter content, high pH, and high calcium carbonate content [13,23]. In the habitats favored by *E. angustifolia*, the water-holding capacity was lower near the topsoil; however, it was higher in the other layers. Therefore, this plant absorbs water from the deeper soil layers and from the groundwater. We did not find any significant correlation with distance from surface water; however, previous studies have confirmed that this plant is associated with surface waters [13]. Rooting depth was lower at the polygons invaded by *E. angustifolia*, so this plant is tolerant of shallow soils. This property makes this invasive plant suitable for the reforestation of degraded areas [23].

The soil organic matter requirement of *R. pseudoacacia* is very low; therefore, it can spread effortlessly in areas where soils are poor in organic matter. As a result, *R. pseudoacacia* can establish dense stands, even in extremely harsh conditions such as those provided in sandy drylands, and by transforming these nutrient-poor areas into nitrogen-rich habitats, it displaces the diverse community of native grassland specialists and creates space for a few weed species [23,24,44]. Since *R. pseudoacacia* is very efficient at taking up organic matter from the soil, the low organic matter content found in invaded areas may be the result of this plant having already exploited these soils. Our results show that *R. pseudoacacia* prefers calcareous soils and favors deeper soils; however, previous studies have found that it is also suitable for reforesting degraded areas [45,46]. In invaded areas, we found that the water-holding capacity near the topsoil was lower, but it was higher in the deeper layers than in non-invaded areas. This suggests that *R. pseudoacacia* absorbs water from the deeper soil layers, even from the groundwater. In relation to climatic conditions, this invasive species prefers warmer and drier habitats; thus, climate change may have a positive impact on its spread [43]. The hydrological results show that *R. pseudoacacia* has a high water demand; however, if the water-holding capacity of the soil is appropriate or the groundwater level is not too low, low precipitation is not a hindrance for this species.

Our results can be used to compare point-based crowd-sourced, and polygon-based distribution maps of the invasive tree species investigated. In our study, we demonstrate the applicability of point- and polygon-based vegetation datasets for the analysis of spatial characteristics of biological invasions and the determination of environmental background variables of invaded areas for a Hungarian case study.

## 5. Conclusions

Our results support the results of previous studies that the occurrence of the investigated invasive tree species is highly dependent on the considered soil properties and hydrological and long-term climatic parameters. Consequently, the effects of these environmental conditions and climate change should be considered when planning the control of these invasive species.

In relation to the usability of the two databases (point- and polygon-based) it was found that the polygon database provides more consistent results on the occurrence of the investigated species using the DOSoReMI database. Polygon-based maps are spatial in extent, and therefore show a better relationship with the DOSoReMI database. The LUCAS point-based database can also be used to determine the relationship between environmental characteristics and the occurrence of invasive plants if sufficient data are available. However, field-based investigations are also highly important to prevent further invasions of these species in the future.

## Figures and Tables

**Figure 1 plants-12-00855-f001:**
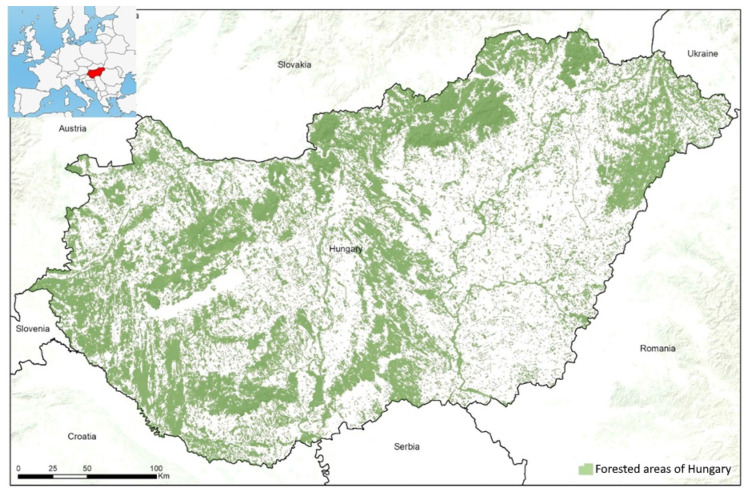
Forested areas (i.e., semi-natural forests, secondary forests, and plantations) of Hungary (based on the National Forestry Database).

**Figure 2 plants-12-00855-f002:**
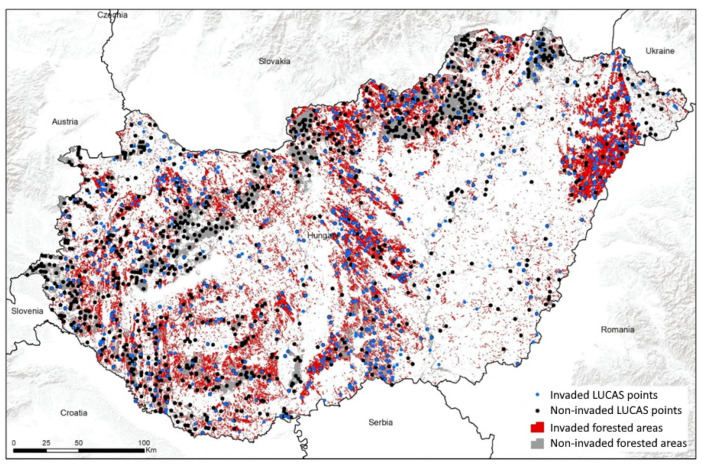
LUCAS points and forest units inside the forested areas invaded by at least one of the three investigated invasive tree species in 2012, 2015, and 2018.

**Table 1 plants-12-00855-t001:** Number of all invaded and non-invaded LUCAS points in the study area for the three investigated tree species in 2012, 2015, and 2018.

	*Ailanthus altissima*	*Elaeagnus angustifolia*	*Robinia pseudoacacia*
Number of all invaded LUCAS points within the total forested area in Hungary	82	38	1086
Number of all non-invaded LUCAS points within the total forested area in Hungary	3349	3389	2345

**Table 2 plants-12-00855-t002:** Percentage of Hungarian forested areas invaded with the three investigated invasive species according to (A) LUCAS point-based data (where 100% = 3432 LUCAS points), (B) NFD forest unit-based occurrence (where 100% = 18, 759 Km^2^), and (C) reference National Forestry Database (2021) statistical data where (where 100% = 18, 759 Km^2^).

	*Ailanthus altissima*	*Elaeagnus angustifolia*	*Robinia pseudoacacia*
(A) Proportions of invaded LUCAS points as a percentage of all LUCAS points inside the total forested area in Hungary	2.29%	1.22%	31.65%
(B) Percentage of NFD polygons invaded with the given species as a percentage of the total forested area in Hungary (2009–2018)	0.02%	0.04%	15.7%
(C) Percentage of areas invaded with the given species as a percentage of the total forested area in Hungary (2021) according to the NFD statistical datasets	0.11%	0.08%	24.48%

**Table 3 plants-12-00855-t003:** Differences between invaded and non-invaded LUCAS points and NFD polygons as a function of the soil, hydrological, and climatic parameters for the invasive tree species investigated.

Investigated Factors	Soil Layers	*Ailanthus altissima*				*Elaeagnus angustifolia*				*Robinia pseudoacacia*			
		LUCAS Points		NFD Polygons		LUCAS Points		NFD Polygons		LUCAS Points		NFD Polygons	
		*p*	z	*p*	z	*p*	z	*p*	z	*p*	z	*p*	z
Organic matter content	0–30 cm	<0.001	−4.092	0.461	−0.737	0.731	0.344	<0.001	6.376	<0.001	−12.737	<0.001	−167.907
	30–60 cm	0.001	−3.215	0.066	1.836	0.018	2.364	<0.001	27.507	<0.001	−6.935	<0.001	−92.761
	60–100 cm	0.072	−1.798	<0.001	6.613	<0.001	3.631	<0.001	39.219	<0.001	−6.296	<0.001	−8.646
	100–200 cm	0.072	−1.798	<0.001	6.597	0.084	1.726	<0.001	24.315	<0.001	−9.579	<0.001	76.131
pH	0–30 cm	<0.001	4.974	<0.001	10.168	<0.001	5.133	<0.001	42.458	<0.001	9.671	<0.001	10.22
	30–60 cm	<0.001	4.701	<0.001	10.769	<0.001	4.942	<0.001	44.516	<0.001	10.478	<0.001	80.276
	60–100 cm	<0.001	4.582	<0.001	10.548	<0.001	4.571	<0.001	43.508	<0.001	11.097	<0.001	96.020
	100–200 cm	<0.001	4.38	<0.001	9.060	<0.001	4.212	<0.001	38.984	<0.001	10.22	<0.001	91.726
Calcium carbonate content	0−30 cm	0.126	1.53	<0.001	7.664	0.080	1.752	<0.001	3.966	0.011	−2.540	<0.001	−71.766
	30–60 cm	0.011	2.532	<0.001	9.848	0.145	1.456	<0.001	5.589	0.377	0.883	<0.001	−54.891
	60–100 cm	0.501	−0.673	<0.001	9.247	0.126	1.529	<0.001	5.589	0.284	−1.071	0.173	1.364
	100–200 cm	0.344	−0.947	<0.001	4.857	0.005	2.794	0.001	3.445	0.001	−3.233	<0.001	−13.393
Rooting depth		<0.001	4.649	0.106	1.619	0.788	−0.268	<0.001	−4.400	<0.001	10.26	<0.001	103.330
Water holding capacity	0–30 cm	<0.001	−4.319	<0.001	−8.034	0.084	−1.726	<0.001	−8.480	<0.001	−8.824	<0.001	−92.671
	30–60 cm	0.002	−3.143	<0.001	−4.098	0.667	0.431	<0.001	14.249	0.636	0.473	<0.001	5.833
	60–100 cm	0.071	−1.806	0.673	0.422	0.023	2.277	<0.001	29.383	<0.001	6.114	<0.001	68.169
	100–200 cm	0.127	−1.525	0.396	0.849	0.032	2.143	<0.001	29.541	<0.001	6.647	<0.001	77.741
Distance from surface water		<0.001	4.779	0.006	−2.732	0.797	−0.258	0.182	1.336	<0.001	6.928	<0.001	−25.649
Groundwater depth		0.338	−0.958	<0.001	−5.633	0.011	−2.531	<0.001	−21.135	0.004	−2.904	<0.001	−58.832
Mean annual temperature		<0.001	3.908	<0.001	6.355	0.004	2.881	<0.001	21.456	<0.001	7.133	<0.001	64.260
Mean annual precipitation		0.010	−2.583	<0.001	−3.707	0.050	−1.96	<0.001	−32.719	<0.001	−10.569	<0.001	−55.131
Legend		The environmental parameters of the invaded points or polygons have significantly higher values than those of the non-invaded points or polygons (at the *p* < 0.05 significance level)											
		The environmental parameters of the invaded points or polygon have significantly higher values than those of the non-invaded points or polygons (at the *p* < 0.001 significance level)											
		The environmental parameters of the invaded points or polygon have significantly lower values than those of the non-invaded points or polygons (at the *p* < 0.05 significance level)											
		The environmental parameters of the invaded points or polygon have significantly lower values than those of the non-invaded points or polygons (at the *p* < 0.001 significance level)											

**Table 4 plants-12-00855-t004:** Soil, hydrological, and climatic factors considered in this study.

	Considered Factor	Soil Layer	Unit of Measurement	Spatial Resolution (Raster Size)
Soil parameters	Organic matter content	0–30 cm	%	100 × 100 m
		30–60 cm		
		60–100 cm		
		100–200 cm		
	pH	0–30 cm	-	100 × 100 m
		30–60 cm		
		60–100 cm		
		100–200 cm		
	Calcium carbonate content	0–30 cm	%	100 × 100 m
		30–60 cm		
		60–100 cm		
		100–200 cm		
	Rooting depth	-	cm	100 × 100 m
	Water holding capacity	0–30 cm	[cm^3^ cm^−3^] × 100	250 × 250 m
		30–60 cm		
		60–100 cm		
		100–200 cm		
Hydrological parameters	Distance from the nearest surface water	-	m	100 × 100 m
	Groundwater depth	-	m	100 × 100 m
Climatic parameters	Mean annual temperature	-	°C	100 × 100 m
	Mean annual precipitation	-	mm	100 × 100 m

## Data Availability

Hungarian GIS Database of Invasive Plant Species http://www.geo.u-szeged.hu/invasive/index_en.html (accessed on 30 December 2022).

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
