# Peer review of "Applicability of Point- and Polygon-Based Vegetation Monitoring Data to Identify Soil, Hydrological and Climatic Driving Forces of Biological Invasions—A Case Study of Ailanthus altissima, Elaeagnus angustifolia and Robinia pseudoacacia"

_plants, 2023, doi:10.3390/plants12040855_

Round 1

Reviewer 1 Report

Review of the paper „Applicability of point and polygon-based vegetation monitoring datas to identify soil, hydrological and climatic driving forces of biological invasions – a case study of Ailanthus altissima, Elaeagnus angustifolia, and Robinia pseudoacacia

The proposed article presents the distribution of three invasive species of shrubs or small trees in Hungary obtained by the method of points and polygons. In addition, environmental data were collected, the impact of which on the distribution of plant invaders was examined through analyzes taking into account points or polygons. Biological invasions are a hot topic, but in the current form of the text, there are no good arguments for choosing such a topic. It is not known how the predicted differences in the influence of environmental factors on the distribution of plant invaders depending on whether they were studied by a point or by polygons may help in the fight against invasions. So, I suggest verifying the concept of work.

Below are detailed suggestions, I am commenting on cited sentences because the lines were not numbered.

Introduction

This chapter is poorly written, especially in its second half. There is a lot of chaos, the aims and questions of the study are mentioned several times in several places. The final sentences in the introduction chapter are short and detached from paragraphs. The paragraphs mostly are not linked with each other.

“Mapping and predicting the occurrence and potential distribution of invasive plants have global significance. Alien species pose a heavy burden on natural ecosystems” – too weak start of the hot topic article. The two sentences are not related to each other. Please, write a few sentences that invasive plants spread dangerously and threaten ecosystems, etc.

„In order to understand the geographical causes of the spread…” – what do you mean by the term “geographical causes”?

“The three alien plants (tree of heaven, Russian olive, black locust), which we will study, show a rapid spread in Eurasian countries.” – starting from this point common names should be supplemented with assigned Latin names, which is customary when full names first appear in the text.

“In this paper, we test how sensitive the relationship between invasive plant occurrence and soil, hydrological and climate conditions is to the type of vegetation map and the type of statistical model used.” – the objectives of the text are repeated in the introduction chapter at least twice. In place of this cited sentence, I would put what are the gaps in knowledge and why it is worth exploring the selected topic. Without it, the introduction seems a bit chaotic, as if the “big question” of the study is missing.

“However, it spreads easily from urban areas and infects other disturbed vegetation and natural forest without closed canopy [15].” – starting from this point I note the phrase “infects”. Infection or infecting are not the appropriate words to refer to invasion. In a citing sentence, it is better to use “invades” and watch for similar words throughout the text.

„We were able to build up country scale point occurrence maps, so called National GIS Database of Invasive Plant Species of Hungary (NDIPS) of the tree investigated invasive tree species, showing the status of infections for each Land Use and Coverage Area frame Survey (LUCAS) field survey point (not infected: there are no invasive species visible, infected: at least some individual plants of a given species are visible on the LUCAS photos).” – rather the status of invasions and (uninvaded…)

“However, the scientific utility of these so-called fragmented big databases (such us Global Biodiversity Information Facility, E-Naturist etc.)” – E-Naturist or i-Naturalist?

Materials and methods

“These GPS recorded (geotagged) field photos from 2012, 2015, and 2018, offer unique possibilities to identify the level of infections of each investigated tree species (Ailanthus altissima, Elaeagnus angustifolia, and Robinia pseudoacacia) and to monitor invasive infections of the whole forest areas of Hungary.” – why are full latin names repeated and no common names? Invasive infections sound like chaos.

“The reliability of the databases was tested by calculating for each species how many invaded LUCAS points fall in the invaded or non-invaded NFD forest unit (polygon) of the given species and how many non-invaded LUCAS points fall in the invaded or non-invaded NFD forest units (polygon). The following formulae were used for the calculations.” – these calculations are incomprehensible and I’m not sure if they are needed in the text.

Results

The caption of Table 2 is also included above in the text.

Table 5 – based on the content of this table alone, it is easy to explain the detectability of the influence of individual environmental factors on the distribution of invasive plants. Examples: in the case of organic matter content, differences between invaded and uninvaded points were detected in shallower soil layers, and in the case of polygons – in deeper ones, because only at the polygon scale the effects associated with deeper locations are detectable, and only what is on the surface is detectable at points. In the case of pH, the differences are similar at points and polygons, because the pH change occurs on a larger scale even than the polygon, and it is difficult to expect that it is different at the point and polygon. Calcium carbonate content differed only in polygons because it is the result of the decomposition of organic matter, the amount of which depends on the plant community. The plant community does not change in points, so it is difficult to expect differences in the point scale. Interestingly, the results are different for the other plant invaders. It is worth discussing in the context of the properties of these plants and the parameters of their invasion, especially the share and role of invasion in the soil and ground.

Conclusions

„The occurrence of the investigated invasive tree species is highly dependent on the considered soil properties, hydrological and long-term climatic parameters. Consequently, these environmental conditions and climate change should be considered when planning the control of these invasive plants.” – as a reader, I am not sure how to use the influence of environmental factors on invasive plants to remove them. I suggest either providing a real way to monitor these factors with tools such as maps, online databases, and results of constant monitoring. The second way is to rewrite the text in such a way that it concerns more invasion science than practice.

Author Response

Thank you very much for evaluating our manuscript and providing us the opportunity to submit again a revised version of the manuscript. Thanks especially for your helpful comments, which drew our attention to the mistakes in the manuscript. I would like to respond to your comments below.

Introduction

This chapter is poorly written, especially in its second half. There is a lot of chaos, the aims and questions of the study are mentioned several times in several places. The final sentences in the introduction chapter are short and detached from paragraphs. The paragraphs mostly are not linked with each other.

Thank you very much for bringing this to our attention. The introduction was indeed difficult to understand and redundant in many places. Based on your comments, we have rethought this section to make it clearer and more explicit. We believe that the research questions were not clearly formulated, so we have also clarified these questions. Please review this section.

“Mapping and predicting the occurrence and potential distribution of invasive plants have global significance. Alien species pose a heavy burden on natural ecosystems” – too weak start of the hot topic article. The two sentences are not related to each other. Please, write a few sentences that invasive plants spread dangerously and threaten ecosystems, etc.

Thank you very much for your comment. The following sentences have been added to this section.

“Mapping and predicting the occurrence and potential distribution of invasive plants is of global significance. Alien species pose a heavy burden on natural ecosystems, displacing native species and transforming natural communities. One of the main issues conservationists face is biological invasion. In addition to natural areas, invasive plants also invade agricultural areas and have a dramatic impact on urban areas, generating enormous extra costs to national economies all over the world.”

„In order to understand the geographical causes of the spread…” – what do you mean by the term “geographical causes”?

For clarification, this section has been replaced by the following:

“To understand the environmental parameters (soil, hydrological and climatic characteristics) determining the distribution of these species, and to model and predict their future spread, up-to-date and detailed spatial data are necessary.”

“The three alien plants (tree of heaven, Russian olive, black locust), which we will study, show a rapid spread in Eurasian countries.” – starting from this point common names should be supplemented with assigned Latin names, which is customary when full names first appear in the text.

Thank you for bringing this to our attention, as it was not consistent. We started by mentioning the English names of the species, then the full Latin names. Hereafter the Latin names have been used in the text, abbreviated as A. altissima, E. angustifolia, R. pseudoacacia. We have chosen to do this because this is how other literature refers to plants in the text.

“In this paper, we test how sensitive the relationship between invasive plant occurrence and soil, hydrological and climate conditions is to the type of vegetation map and the type of statistical model used.” – the objectives of the text are repeated in the introduction chapter at least twice. In place of this cited sentence, I would put what are the gaps in knowledge and why it is worth exploring the selected topic. Without it, the introduction seems a bit chaotic, as if the “big question” of the study is missing.

Thank you for your helpful comment, the quoted sentence has been replaced with the following:

“To be able to use these databases for modelling hazard mapping, we need to determine which types of database results provide the most reliable information.”

“However, it spreads easily from urban areas and infects other disturbed vegetation and natural forest without closed canopy [15].” – starting from this point I note the phrase “infects”. Infection or infecting are not the appropriate words to refer to invasion. In a citing sentence, it is better to use “invades” and watch for similar words throughout the text.

We fully agree with you, we have corrected the incorrect phrase "infects" to "invades" in the whole text.

„We were able to build up country scale point occurrence maps, so called National GIS Database of Invasive Plant Species of Hungary (NDIPS) of the tree investigated invasive tree species, showing the status of infections for each Land Use and Coverage Area frame Survey (LUCAS) field survey point (not infected: there are no invasive species visible, infected: at least some individual plants of a given species are visible on the LUCAS photos).” – rather the status of invasions and (uninvaded…)

We would like to thank you for noticing this mistake. The term not infected has been corrected to uninvaded.

“However, the scientific utility of these so-called fragmented big databases (such us Global Biodiversity Information Facility, E-Naturist etc.)” – E-Naturist or i-Naturalist?

Thank you for bringing this to our attention. The incorrect “E-Naturist” placement has been replaced by the correct “iNaturalist”.

Materials and methods

“These GPS recorded (geotagged) field photos from 2012, 2015, and 2018, offer unique possibilities to identify the level of infections of each investigated tree species (Ailanthus altissima, Elaeagnus angustifolia, and Robinia pseudoacacia) and to monitor invasive infections of the whole forest areas of Hungary.” – why are full latin names repeated and no common names? Invasive infections sound like chaos.

From this part of the text, we have deleted the names of the species, because we mentioned them earlier. The term “invasive infections” has been corrected to infection. Thank you for noticing this serious mistake.

“The reliability of the databases was tested by calculating for each species how many invaded LUCAS points fall in the invaded or non-invaded NFD forest unit (polygon) of the given species and how many non-invaded LUCAS points fall in the invaded or non-invaded NFD forest units (polygon). The following formulae were used for the calculations.” – these calculations are incomprehensible and I’m not sure if they are needed in the text.

These calculations do not present a new result, but the quantified values based on the two databases. However, this is necessary to understand what the research was based on. Since these calculations are not really the core of the research, the formulas and the corresponding table in the results section have been moved to the appendix.

Results

The caption of Table 2 is also included above in the text.

The text in Table 2 has been corrected to read: “Number of all invaded and non-invaded LUCAS points in the study area for the three investigated tree species in 2012, 2015, and 2018.”

Table 5 – based on the content of this table alone, it is easy to explain the detectability of the influence of individual environmental factors on the distribution of invasive plants. Examples: in the case of organic matter content, differences between invaded and uninvaded points were detected in shallower soil layers, and in the case of polygons – in deeper ones, because only at the polygon scale the effects associated with deeper locations are detectable, and only what is on the surface is detectable at points. In the case of pH, the differences are similar at points and polygons, because the pH change occurs on a larger scale even than the polygon, and it is difficult to expect that it is different at the point and polygon. Calcium carbonate content differed only in polygons because it is the result of the decomposition of organic matter, the amount of which depends on the plant community. The plant community does not change in points, so it is difficult to expect differences in the point scale. Interestingly, the results are different for the other plant invaders. It is worth discussing in the context of the properties of these plants and the parameters of their invasion, especially the share and role of invasion in the soil and ground.

Conclusions

„The occurrence of the investigated invasive tree species is highly dependent on the considered soil properties, hydrological and long-term climatic parameters. Consequently, these environmental conditions and climate change should be considered when planning the control of these invasive plants.” – as a reader, I am not sure how to use the influence of environmental factors on invasive plants to remove them. I suggest either providing a real way to monitor these factors with tools such as maps, online databases, and results of constant monitoring. The second way is to rewrite the text in such a way that it concerns more invasion science than practice.

The databases used contain both 100*100 m soil data and therefore offer the possibility to compare the soils under the points and polygons. The aim of our study was not to compare the variability of the soil beneath the points and polygons, but to compare the relationship between the occurrence of plants and parameters based on different databases. In this research, the aim was to produce input data for modelling and hazard mapping. To do this, such basic research is essential to show the impact of environmental parameters on the spread of invasive plants.

Reviewer 2 Report

The authors present an interesting study dealing with one of the main threats of the present global change scenario that is the invasive tree (and other) species. Understanding the biology of the invasive and the mechanisms of early fight allow us to perform better management on early stages of the invasive. In this sense the study is very interesting.

The authors focus the study in top-three invasive trees in Hungary and using GIS and different data sources could analyses the correlation with environmental variables.

The manuscript is well written, the methodology is clear and proper, the exposition of results is adequate and fit with the objectives of the study. The extent and quality of the bibliography is appropriate.

The manuscript is ready for publication after minor changes that which I explain below.

1.      English. I recommend an in-depth analysis in order a better comprehension of the whole manuscript.

2.      In introduction section a couple of sentences are needed to connect the presentation of your research with the ecology of the invasive species.

3.      Review the acronyms please. Sometimes appears MFD, NFD, NDF polygon-based occurrence,…. Some kind of uniformity is needed specially when there are many acronyms in the manuscript.

4.      Review the caption of Table 3. Where 100%=3432 what unit ?

5.      Tables 5, 6 and 7 are very interesting because synthesizes your results. I suggest join the three tables in one (doesn’t matter that it is big) and to clarify the legend. Under my humble opinion the legend should improve because there is a little trouble with colors (blue and grey colors).

Congratulations for the research. I’ve enjoy reading your study.

Author Response

Thank you very much for evaluating our manuscript and providing us the opportunity to submit again a revised version of the manuscript. Special thanks to you for looking over our manuscript so carefully and noticing small, but still very significant errors, such as the incorrect use of acronyms in the text.

Below I describe the changes I made based on your helpful suggestions.

  1. I recommend an in-depth analysis in order a better comprehension of the whole manuscript.

      Thank you very much for your helpful feedback. In order to avoid problems with the English language of the manuscript, we have proofread the English text.  

  1. In introduction section a couple of sentences are needed to connect the presentation of your research with the ecology of the invasive species.

The introduction has been rethought to make it easier to understand. The following sentence is used to introduce the plants examined:

“In this study, we investigated the occurrence of three invasive tree species (Ailanthus altissima, Elaeagnus angustifolia and Robinia pseudoacacia). Among woody plants, these species present the highest risk of biological invasion in Hungary.

Review the acronyms please. Sometimes appears MFD, NFD, NDF polygon-based occurrence,…. Some kind of uniformity is needed specially when there are many acronyms in the manuscript.

      Thank you very much for your comment, the acronyms were incorrect in several places. The correct acronym is NFD (National Forestry Database).

      The acronyms in five places have been corrected.

  1. Review the caption of Table 3. Where 100%=3432 what unit ?

      Thank you for noticing this mistake, it was indeed not clear what it was about. We have made the following addition: “(Where 100%=3432 LUCAS points)”.

  1. Tables 5, 6 and 7 are very interesting because synthesizes your results. I suggest join the three tables in one (doesn’t matter that it is big) and to clarify the legend. Under my humble opinion the legend should improve because there is a little trouble with colors (blue and grey colors).

      Thank you very much for your kind observation, I absolutely agree. Instead of three tables, we made one table showing the results for all three plants. In addition, the legend has been improved and the colors have been standardised.

Reviewer 3 Report

The work submitted for review describes the serious problem of invasive trees that is noted throughout Europe, where species from Asia as well as North America are spreading. Monitoring as well as conducting research on these taxa is very important and vital for the European flora. I would also suggest supplementing the literature review with the response of hogweed to the heat island in large cities in central Europe such as WrocÅ‚aw.  In Poland, where Ailanthus altissima has spread in large cities, and most of its stands have been recorded in the city center, where the urban heat island takes place.  In contrast, for Robinia pseudoacacia, it is common in most areas in Poland's cities and does not respond to the urban heat island. Eleagnus angustifolia is not an invasive species in Poland.  I believe that the introduction of Robini pseudoacacia as a forest species was a serious mistake, as forests should be established only with native tree and shrub species.
What is interesting in the presented article is the link from the databases of studies on the distribution of invasive taxa to map data in comparison with important hydrological and soil and climatic factors such as average temperature.

Author Response

Thank you very much for evaluating our manuscript and providing us the opportunity to submit again a revised version of the manuscript. Thank you especially for suggesting some additions to the literature that were missing from the text but are an important part of understanding the problem. This is very important because invasive species not only threaten protected areas, but also cause serious economic damage and can have a negative impact on the urban environment, such as Ailanthus altissima. And the costs of preventing their spread and removing them are significant.

Below I describe the changes I made based on your helpful suggestions.

In the introduction, we mentioned that Ailanthus altissima is concentrated on urban heat islands and is particularly widespread in urban environments. The fact that air temperatures in urban heat islands are a few degrees higher suggests that this species may benefits from higher temperatures. For this reason, we have mentioned in the discussion that climate change may favour the spread of this species.

You can find the relevant sentences on lines 80-85 in the introduction and lines 344-347 in the discussion.

The following literature has been added:

  1. Paź-Dyderska, S.; Ladach-Zajdler, A.; Jagodziński, A.M.; Dyderski, M.K. Landscape and Parental Tree Availability Drive Spread of Ailanthus Altissima in the Urban Ecosystem of Poznań, Poland. Urban For Urban Green 2020, 56, doi:10.1016/j.ufug.2020.126868.

  1. Przemysław Bbelewski, M.C. Distribution of Tree of Heaven, Ailanthus Altissima (Mill.) Swingle, in Wrocław, Lower Silesia, Poland 2005

I totally agree that forests should only be planted with native species, but the economic benefits of acacia are so significant that I think there is no way they will stop planting it.

Round 2

Reviewer 1 Report

I appreciate the work effort made by authors. The paper was improved, the content is interesting. Now the paper is publishable.